# Proteomic Profiling of Extracellular Vesicles Isolated from Plasma and Peritoneal Exudate in Mice Induced by *Crotalus scutulatus scutulatus* Crude Venom and Its Purified Cysteine-Rich Secretory Protein (Css-CRiSP)

**DOI:** 10.3390/toxins15070434

**Published:** 2023-07-02

**Authors:** Armando Reyes, Joseph D. Hatcher, Emelyn Salazar, Jacob Galan, Anton Iliuk, Elda E. Sanchez, Montamas Suntravat

**Affiliations:** 1National Natural Toxins Research Center (NNTRC), Texas A&M University-Kingsville, MSC 224, 975 West Avenue B, Kingsville, TX 78363, USA; armando.reyes@students.tamuk.edu (A.R.); joseph.hatcher@students.tamuk.edu (J.D.H.); emelyn.salazarcastillo@tamuk.edu (E.S.); elda.sanchez@tamuk.edu (E.E.S.); 2Department of Human Genetics, The University of Texas Rio Grande Valley School of Medicine, Brownsville, TX 78539, USA; jacob.galan@utrgv.edu; 3Tymora Analytical Operations, West Lafayette, IN 47906, USA; anton.iliuk@tymora-analytical.com; 4Department of Chemistry, Texas A&M University-Kingsville, MSC 161, Kingsville, TX 78363, USA

**Keywords:** snakebites, vascular permeability, snake venom cysteine-rich secretory proteins (svCRiSPs), extracellular vehicles (EVs), proteomic analysis

## Abstract

Increased vascular permeability is a frequent outcome of viperid snakebite envenomation, leading to local and systemic complications. We reported that snake venom cysteine-rich secretory proteins (svCRiSPs) from North American pit vipers increase vascular permeability both in vitro and in vivo. They also induce acute activation of several adhesion and signaling molecules that may play a critical role in the pathophysiology of snakebites. Extracellular vesicles (EVs) have gained interest for their diverse functions in intercellular communication, regulating cellular processes, blood-endothelium interactions, vascular permeability, and immune modulation. They also hold potential as valuable biomarkers for diagnosing, predicting, and monitoring therapeutic responses in different diseases. This study aimed to identify proteins in peritoneal exudate and plasma EVs isolated from BALB/c mice following a 30 min post-injection of *Crotalus scutulatus scutulatus* venom and its purified CRiSP (Css-CRiSP). EVs were isolated from these biofluids using the EVtrap method. Proteomic analysis of exudate- and plasma-derived EVs was performed using LC-MS/MS. We observed significant upregulation or downregulation of proteins involved in cell adhesion, cytoskeleton rearrangement, signal transduction, immune responses, and vesicle-mediated transports. These findings suggest that svCRiSPs play a crucial role in the acute effects of venom and contribute to the local and systemic toxicity of snakebites.

## 1. Introduction

The array of toxins in viperid snakes can induce both local and systemic effects that are clinically heterogeneous. Interestingly enough, more than 90% of viper venoms are composed of only ten main protein families, including phospholipase A_2_s (PLA_2_), snake venom metalloproteases (SVMPs), serine proteases (SVSPs), three-finger toxins (3FTxs), cysteine-rich secretory proteins (CRiSPs), disintegrins (DIS), C-type lectins/snaclecs (CTLs), L-amino acid oxidases (LAAOs), kunitz peptides, and natriuretic peptides [1]. These protein families are the cause of most severe symptoms post-envenomation. One of the most common responses to snake venom is increased vascular permeability, leading to local complications, such as pain, edema, and swelling, followed by serious systemic complications, including inflammation, tissue damage, shock, organ failure, and even death [2]. Snake venom toxins known to induce vascular permeability include snake venom vascular endothelial growth factors (svVEGFs) [3,4], SVMPs [5,6], PLA_2_s [7], CTLs [8,9,10], and CRiSPs [11,12]. Among them, CRiSPs are relatively understudied, even though they are being explored among different snake venoms. These proteins have various biological functions, including ion channel inhibition, anti-angiogenic activity, vascular permeability regulator, and promoting inflammatory responses and protease activity [11,13]. Our previous study showed that crotaline CRiSPs alter key components of cellular signaling pathways, promoting endothelial dysfunction, and can contribute significantly to the pathophysiology of snake envenomation [12].

Investigating snake envenoming-induced alterations in the host proteome can provide valuable information regarding molecular biological features of the complex interactions between venom components and host physiology. Several studies have utilized proteomic techniques to identify the novel potential biomarkers of tissue alterations for understanding the pathogenesis of local effects induced by *Bothrops asper* venom and some of its toxins [14,15,16,17,18]. Cavalcante et al. have recently investigated and identified the systemic pathological and inflammatory events at the early alterations in the plasma proteome of mice envenomated by *B. atrox* venom [19,20]. However, proteomic analysis of blood plasma is challenging due to the presence of highly abundant proteins, such as albumin and immunoglobulins, which can interfere with the detection of low-abundance proteins that may be relevant to the disease [21]. Recent studies reported that proteome analysis of extracellular vesicles (EVs) might allow for improved detection and quantitation of disease-specific proteins in low concentrations [21,22,23].

EVs are nanosized lipid-bilayer vesicles released by most cell types into biological fluids [24]. They are key players in intercellular communication and the transport of biological molecules, including nucleic acids, proteins, metabolites, and lipids [25]. They are relatively stable and can be isolated and stored for long periods without degradation [26,27]. Proteomic profiling of EVs and their contents has shown significant alterations in protein expression under various pathophysiological conditions [28,29,30], contributing to a better understanding of disease pathogenesis, progression, and severity and identification of novel biomarkers for various diseases and conditions, such as cancers, cardiovascular diseases, autoimmune diseases, infectious diseases, and vascular permeability [31,32,33,34]. Recently, our proteomic analysis of EVs isolated from the plasma of mice injected with *Crotalus atrox* and *C. oreganus helleri* has revealed quantified potential biomarkers for late systemic effects of snake envenomation [35]. However, the pathophysiology of an early stage of snake envenomation is yet to be fully understood. The exact pathophysiology of snake envenomation varies depending on the type of snake and the specific venom components involved [36].

To our knowledge, no study has been performed to identify EVs in post-envenomation biofluids from mice injected with crude venom of *C. s. scutulatus* and its purified CRiSP (Css-CRiSP). The aim is to characterize EVs from the peritoneal exudate and plasma and, more specifically, identify target proteins involved in the local and systemic effects of early pathophysiology of snake envenoming. Therefore, we investigated the proteome expression profile of EVs isolated from biofluid samples from mice at 30 min after injection with crude venom and Css-CRiSP using a label-free quantitation (LFQ) LC-MS/MS method. Considering that CRiSPs appear to be relevant for the acute local effects of crotalid snake envenoming, especially vascular barrier function, the identification of EVs from biofluids of envenomated mice with snake venom and CRiSP utilizing a proteomic approach can offer a better understanding of the mechanism of action of this toxin and the complex molecular interactions at early pathogenesis events in snakebite and inform the development of new therapeutic strategies.

## 2. Results

### 2.1. Comparison of Peritoneal Exudate- and Plasma-Derived EV Protein Cargo with the Exocarta and Vesiclepedia Databases

In this study, we aimed to gain insight into the role of Css-CRiSP in *C. s. scutulatus* envenomation by characterizing the proteomic profiles in peritoneal exudate-derived EVs (E-EV) and plasma-derived EVs (P-EV) from mice with acute administration of Css-CRiSP and crude venom. EVs were isolated from exudate and plasma samples using the EVtrap method and analyzed using high-resolution LC–MS/MS. The label-free quantitative proteomics analysis identified a total of 3216 proteins from exudate EVs (E-EV) (Appendix A) and 3091 proteins from plasma EVs (P-EV) (Appendix A). Both sets of EVs had 2153 proteins (51.8%) in common, 1063 were specific to EVs from exudates, and 938 were specific to EVs from plasma (Figure 1A), indicating the difference in protein components in EVs derived from different biofluids.

To ensure the enrichment of our EV samples isolated from EVTrap, we compared qualitatively the proteins identified in E-EV and P-EV to the top 100 exosomal/EV markers using public databases, including Exocarta and Vesiclepedia [37,38]. As a result, we found more than 60% of the top 100 EV proteins from the databases in our P-EVs and E-EVs (Figure 1B,C, Appendix A). These results demonstrated that the samples were enriched with known EV markers after isolation. Furthermore, the similarities between conjunctions in databases confirm that we isolated EVs from the respective biofluids.

### 2.2. Proteomic Analysis of Peritoneal Exudate-Derived EVs from Mice Envenomated with Crude Venom (E-vEV) and Css-CRiSP (E-CRiSP-EV)

We performed quantitative proteomic analysis to characterize the protein content of peritoneal exudate EVs from mice injected with *C. s. scutulatus* venom (E-vEV) or Css-CRiSP (E-CRiSP-EV) compared to the control group (P-cEV). We identified 2823 proteins, of which 2331 were overlapped in all groups, 8 were unique to E-vEV, 37 were specific to E-CRiSP-EV, and 136 were specific to E-cEV (Figure 2A). The proteins expressed in all groups were quantified, hierarchically clustered, and represented in a heatmap, as shown in Figure 2B. We then performed the volcano scatter plot of upregulated and downregulated proteins in E-vEV (Figure 2C) and E-CRiSP-EV (Figure 2D). Additionally, we found a total of 2455 quantified proteins in E-vEV (Figure 2E), while 2476 proteins were quantified in E-CRiSP-EV (Figure 2F). To explore the differences in expression between the E-vEV and E-CRiSP-EV, we performed the normalized label-free quantitative proteomics. We set the fold change (FC) using the thresholds of ±2FC in E-vEV and E-CRiSP-EV over the control (*p* < 0.05). Compared with the control group, 965 differentially expressed proteins were identified in the E-vEV, 241 were upregulated, and 724 were downregulated (Figure 2E, Appendix A). Additionally, we identified 721 differentially expressed proteins, including 339 upregulated and 382 downregulated proteins in E-CRiSP-EV (Figure 2F, Appendix A).

We have previously demonstrated that CRiSPs induce acute vascular permeability and activate signaling pathways that may contribute to the pathophysiology of a snakebite [11,12]. To gain insights into the cellular functions and biological processes of CRiSP in the early stage of snake envenoming, we selected proteins that significantly changed (±2 FC, *p* < 0.05) in both E-vEV and E-CRiSP-EV (Appendix A) and performed a gene ontology (GO) enrichment analysis. As a result, we found 364 differentially expressed proteins (124 upregulated and 240 downregulated) in both E-vEV and E-CRiSP-EV compared to the control (Figure 3). In addition, we found that upregulated proteins in E-vEV and E-CRiSP-EV were significantly enriched with GO categories linking to complement activation, B cell receptor signaling pathway, phagocytosis, and innate immune response. In contrast, the downregulated proteins were associated with proton transport, ATP metabolic process, mitochondrial membrane, and focal adhesion (Figure 3B). Top GO annotation terms from enrichment analyses are presented in Appendix A and top significant functional annotation clusters are presented in Appendix A.

We also performed a protein–protein interaction analysis using the STRING online database to investigate the interaction patterns of proteins identified in E-EV and E-CRiSP-EV (Figure 4). STRING analysis revealed that upregulated proteins are involved in the innate immune response, complement coagulation cascades, and cell junction. We also identified downregulated proteins involved in the mitochondria, metabolic pathways, focal adhesion, actin cytoskeleton organization, and endoplasmic reticulum (ER).

### 2.3. Proteomic Analysis of Plasma-Derived EVs from Mice Envenomated with Crude Venom (P-vEV) and Css-CRiSP (P-CRiSP-EV)

To characterize the proteins and analyze the possible immediate systemic responses of envenomated mice, we performed a quantitative proteomic analysis of plasma EVs from mice injected with *C. s. scutulatus* venom (P-vEV) or Css-CRiSP (P-CRiSP-EV) compared to the control group (P-cEV). The label-free proteomics analysis identified and quantified 2602, 2554, and 2551 proteins in P-vEV, P-CRiSP-EV, and P-cEV, respectively (Figure 5A). This comparison revealed 2462 (93%) coinciding proteins within all three groups, as illustrated in a Venn diagram (Figure 5A). We also identified a large number of proteins (2519 proteins, 95%) that were found to be common between P-vEV and P-CRiSP-EV (Figure 5A). We then performed unsupervised hierarchical clustering to identify the changes in expression levels of proteins between P-vEV and P-CRiSP-EV compared to P-cEV (Figure 5B). The heatmap shows protein expression patterns and clustering between P-vEV and P-CRiSP-EV compared to the control (P-cEV). In addition, the volcano plot indicated the differentially expressed proteins between P-vEV (Figure 5C) and P-CRiSP-EV (Figure 5D). The volcano plots and the heat map were generated using the log2 transformed FC values. The upregulated and downregulated proteins are represented by the red and blue colors, respectively.

To analyze the difference in protein expressions, we determined changes in expression levels of proteins in P-vEV and P-CRiSP-EV using 2-fold for upregulated and 0.5-fold for downregulated proteins (*p* < 0.05). In the label-free quantitation method, 2515 proteins were quantified from P-vEV, with 239 upregulated proteins and 104 downregulated proteins compared to P-cEV (Figure 5E, Appendix A). The analysis of P-CRiSP-EV revealed 2409 quantifiable proteins, with 126 upregulated proteins and 114 downregulated proteins (Figure 5F, Appendix A).

To analyze whether Css-CRiSP has an effect that contributes to the pathophysiology of snakebites, we selected 86 proteins that changed more than 2-fold in both P-vEV and P-CRiSP-EV (Appendix A). Of these, 75 proteins were significantly upregulated (FC > 2, *p* < 0.05), and 11 were downregulated (FC < 0.5, *p* < 0.05). We also generated an unsupervised hierarchical heatmap to visualize protein expression changes in P-vEV and P-CRiSP-EV compared to the control (Figure 6A). There is a consistent expression pattern and clustering between P-vEV and P-CRiSP-EV, indicating the significant role of CRiSP at the early stage of snakebite.

To identify the key signaling pathways altered in the early phase of snake envenoming in both P-vEV and P-CRiSP-EV, the selected proteins were then analyzed by clustering and GO enrichment analysis. GO analysis focusing on these 75 proteins that were increased in both P-vEV and P-CRiSP-EV showed a significant enrichment of proteins involved in protein homeostasis, metabolic pathways, signaling pathways, and intracellular protein transport (Figure 6B). Top GO annotation terms from enrichment analyses are presented in Appendix A, and the top significant functional annotation clusters are shown in Appendix A. Notably, no significant enrichment of GO terms was found in downregulated proteins. Next, we used the database provided by STRING to predict protein–protein interactions. The bioinformatics software revealed that the significantly upregulated proteins were involved in the proteasome, cytosol, MAPK6/MAPK4 signaling, TNF-α, nuclear factor-kappa-B (NF-kB) signaling pathway, metabolism, and innate immune system (Figure 6C).

### 2.4. Comparison of vEV and CRiSP-EV Isolated from Mice Plasma and Peritoneal Exudate

Hierarchical clustering analysis of the vEV and CRiSP-EV illustrates the differences in the expression levels of these identified proteins isolated from mice exudate (Figure 3A) and plasma (Figure 6A). Most of these changes demonstrate the downregulation of protein expression in the exudate, while the upregulation of protein expression is observed in the plasma.

To obtain a better insight into the role of svCRiSPs in crude venom and analyze the protein profiles of EVs from both biofluids in more detail, we compared the expression of the top ten significantly upregulated or downregulated proteins expressed in vEV and CRiSP-EV from mice injected with crude venom and the equipotent sublethal dose of Css-CRiSP. Figure 7 shows the expression of the top 10 upregulated and downregulated altered proteins in vEV and CRiSP-EV from exudate and plasma with log2 FC of 1 and −1, *p* < 0.05, compared to control mice and their biological processes determined by GO are presented in Appendix A. We analyzed the major biological functions of the proteins listed using DAVID mouse data resources. The most upregulated proteins found in both E-vEV and E-CRiSP-EV (Appendix A) are known to affect protein transport, innate immunity, mRNA splicing, and inflammatory response. On the other hand, proteins that were least expressed in E-EVs are involved in I-kappaB kinase (IKK)/NF-kB signaling, actin cytoskeleton organization, cell–cell junction organization, mitochondrion organization (Appendix A). Furthermore, we observed the most common upregulation of proteins in both P-vEV and P-CRiSP-EV related to complement activation, platelet aggregation, ER to Golgi vesicle-mediated transport, and IKK/NF-kB signaling (Appendix A). We also observed the most common downregulation of these P-EVs related to cell adhesion, cytoskeleton organization, tricarboxylic acid cycle, and mitochondrial electron transport (Appendix A).

This comparison revealed a consistency in the over-expression and suppression of similar proteins in both treatments (vEV and CRiSP-EV) in exudate and plasma, indicating that Css-CRiSP could potentially be involved in the immediate effects of the envenomation.

## 3. Discussion

Proteomic analysis is a well-established approach for identifying EV proteins involved in the development and progression of various diseases, including neurodegenerative diseases, cancers, injury, local and systemic inflammation, and vascular pathophysiology [39,40,41,42,43,44,45,46,47,48]. Identification of EV protein candidates can help discover novel EV-based biomarkers for diagnosis, prognosis, and monitoring of disease progression.

Our previous proteomic studies focused on characterizing the EVs in the systemic circulation 48 h after the injection of crude venoms [35], indicating its application to the study of physiological changes in the envenomated model. In addition, we have demonstrated the important role of svCRiSPs from North American rattlesnakes in the acute effect on vascular and endothelial permeability both in vitro and in vivo, regulation of endothelial permeability, and modulation of key signaling pathways [11,12]. Because we found a significant effect of svCRiSP, here we investigated the proteomic profiling of vEV and CRiSP-EV isolated from exudate and plasma to better understand possible local and systemic effects of svCRiSPs at the early stage of snake envenomation.

Plasma or serum is the most used biofluid source to isolate EVs to identify specific proteins produced in the systemic response of diseases, drug treatments, and other interventions [39,40]. Recently, it was shown that exudate associated with wounds is a rich source of molecules and has recently been used to study the local tissue damage induced by snake venoms [14,15,16,17,18,49]. Moreover, several studies have shown that peritoneal exudate-derived EVs contain many EV biomarkers related to the local response to inflammation and disease progression [50,51,52,53]. However, isolating EVs from a biofluid is challenging with regard to yield, reproducibility, and purity [54]. To overcome this, we implemented the recently developed EVtrap method, a technique designed to capture membrane-bound vesicles, including exosomes. It has been used in several studies to isolate EVs for various applications, including the study of the development and/or progression of many diseases and disease biomarkers, due to its simplicity, efficiency, and versatility [55,56,57,58,59]. This is the first study to isolate and identify EV markers from peritoneal exudate from mice envenomated with snake venom and its toxins.

In this study, we performed proteomic profiling of vEV and CRiSP-EV from mouse peritoneal exudate and plasma 30 min after envenomation. We demonstrated that about 52% of proteins identified in E-EV are common in P-EV, validating the use of different biofluids as sources of EVs (Figure 1A) [60]. Furthermore, the P-EV and E-EV contained approximately 60 of the top 100 EV proteins in Vesiclepedia and Exocarta databases (Figure 1B). These results indicate that EVs were successfully isolated from mouse peritoneal exudate and plasma.

The proteomic profiling of vEV and CRiSP-EV from exudate and plasma was performed to elucidate the immediate effects of snake envenomation locally and systemically, mainly focusing on the role of CRiSP. We demonstrated that most proteins are common in vEV, CRiSP-EV, and cEV from exudate (83%, Figure 2A) and plasma (93%, Figure 5A). Differentially expressed proteins were initially screened out among these detected proteins according to the FC ≥ 2. Interestingly, there were 965 (41%) and 721 (31%) proteins significantly different from E-vEV and E-CRiSP-EV, respectively (Figure 2E,F), while 343 (14%) and 240 (10%) were significantly altered in P-vEV and P-CRiSP-EV, respectively (Figure 5E,F). The slightly greater significant changes of E-EVs and P-EVs from mice treated with crude venom than those treated with the equipotent sublethal dose of CRiSP may be due to the complexity of the venom mixture that can have a diverse range of effects on cells, tissues, and immune response compared to purified CRiSP alone.

Viperine CRiSPs have been previously shown to induce vascular permeability, activate key components of cellular signaling pathways, and promote inflammatory responses [11,12,13]. Therefore, it is reasonable to hypothesize that Css-CRiSP may play an important role in the early stage of Mojave snakebite effects by inducing intracellular signaling that regulates specific pathways and controls biological responses. We selected the common significant changes in both vEV and CRiSP-EV from exudate and plasma (FC > 2, *p* < 0.05). Upon comparing the hierarchical clustering of significantly expressed proteins in exudate and plasma, it was observed that most of the proteins identified in exudate EVs were downregulated, while the EVs in plasma were enriched with upregulated proteins (Figure 3A and Figure 6A). The data can uncover distinct mechanisms underlying the acute effects of crude venom and toxin and the host’s defense against envenomation, in both local and systemic responses.

We conducted the DAVID GO analysis based on biological process, cellular component, and molecular function implicated to delineate the categories of proteins in E-EVs and P-EVs. The upregulated proteins in E-vEV and E-CRiSP-EV were shown to be involved in innate immune responses, complement activation, phagocytosis, and the immune system (Figure 3B). The downregulated proteins were significantly enriched in lipid and ATP metabolism processes, mitochondrion, ER membrane, cytoskeleton, and actin and cell adhesion binding. As shown by the protein network and pathway analysis (Figure 4), the major proteins are implicated in response to immune responses, proteosomes, MAP kinase pathways, cell junctions, and cytoskeletons. Our results are in accordance with previous reports on the mechanisms related to inflammation and hemostatic alterations induced by intraperitoneal injection of viperid venoms or toxins (such as SVMPs and PLA2s) that can cause the alteration of adhesion molecules, inflammatory mediators, lipid-derived mediators, vascular leakage, and amplifying Viperidae envenomation [61,62,63,64,65,66,67,68]. Additionally, the GO analysis and network analysis of P-EVs illustrated a systemic effect triggered by crude venom and CRiSP with the upregulation of proteins involved in intracellular protein transport, protein catabolic process, metabolic processes, homeostatic process, and the inflammatory signaling pathways (Figure 6B,C). This observation was consistent with the changes observed in the mouse plasma proteome 30 min after *B. atrox* injection [19].

We further evaluated the most upregulated proteins (Exoc8 and Apoa2) in both E-vEV and E-CRiSP-EV (Figure 6). Exoc8 (also known as Exo84 or Exo84p) was upregulated the most in E-EVs. This protein is a component of the exocyst complex that plays a critical role in the assembly of the exocyst complex, exocytosis of lysosomes or intracellular vesicles, vesicular trafficking, the secretory pathway by targeting post-Golgi vesicles to the plasma membrane, and membrane repair [69,70,71,72,73]. Exocytosis is one of the methods of membrane repair that frequently occurs in vivo during cell membrane injuries, especially large injuries [74]. This finding agrees with our previous studies reporting the effect of svCRiSPs on acute endothelial permeability and vascular leakage [11,12], suggesting that Exoc8 is essential to repair these membrane disruptions induced by svCRiSP and might also serve as an early envenomation marker.

Apolipoprotein A-II (ApoA-II or Apoa2) represents a significant component of high-density lipoprotein (HDL), serving as the second most prominent apolipoprotein [75]. Several studies revealed the overexpression of ApoA-II is involved in inflammation, lipid mechanism, and oxidative stress markers [76,77,78,79,80]. Recently, it has been shown that Apoa2 knockout mice had less tissue damage and less inflammatory cell infiltration during acute-phase response and may play an important role in the pathogenesis of amyloid A amyloidosis in mice [81]. Collectively, and considering the observed upregulation of Apoa1 and Apoa2 in mice plasma 30 min after the injection of *Bothrops erythromelas* crude venom [82], further investigation of Apoa2 as a potential systemic inflammation marker in snakebite is suggested.

In addition to upregulated proteins in E-EVs, many downregulated proteins were also involved in the acute effects of snake envenoming. The common significantly downregulated proteins in both E-vEV and E-CRiSPs-EV were Cnn3, Dhx36, Smc3, and Ppa1. Calponin-3 (Cnn3) is a member of the calponin family comprising actin filament-associated proteins [83]. It can bind to binding to F-actin, calmodulin, and tropomyosin and regulates actin cytoskeleton reorganization and dynamics [84]. A decrease in Cnn3 expression was shown to promote actin cytoskeletal rearrangement, contractility, plasticity, fibrogenic activity, and mechanosensitive Yap/Taz transcriptional activation [85,86]. The alteration of cytoskeleton dynamics appears to be the first target during endothelial barrier disruption induced by agonists such as thrombin and barrier-disruptive agents [11,87]. Consistent with our previous study Hellerin, a CRiSP isolated from *C. o. helleri*, induced the reduction in the F-actin cytoskeleton in human endothelial cells [11].

DEAH-box polypeptide 36 (Dhx36) is also known as RNA helicase and is involved in maintaining genomic integrity and initiating the proper transcription and translation [88]. Dhx36 is also a critical component in immunity and the interferon (IFN) signaling pathway [89]. Dhx36 knockdown is associated with a significant decrease in virus-triggered double-stranded RNA-activated protein kinase R phosphorylation and IFN-I production, suggesting its host cells’ antiviral innate immune responses [90]. Smc3 is a cohesin complex component with canonical roles in cell cycle progression and checkpoint control. Smc3 deficiency in mammalian cells affects tissue morphogenesis leading to the activation of an apoptotic cascade involving p53 [91]. Furthermore, the downregulation of Smc3 in influenza A virus-infected lung epithelial cells and in damaged axons was observed [92,93], suggesting that a decrease in SMC3 is a part of the response to virus infection and other cellular stress. Lastly, inorganic pyrophosphatase 1 (Ppa1) is an important enzyme in cell growth, cellular metabolism, energy metabolism, and PI3K/Akt signaling pathway [94]. It was found that the downregulation of Ppa1 induces apoptosis by increasing p53 and decreasing β-catenin levels [95]. The downregulation of these proteins may have reflected an innate immune response to acute local effects induced by svCRiSPs.

The top abundant proteins in both P-vEV and P-CRiSP-EV were Ighv5-17, Myo18a, Gcdh, and Pdcl3. Ighv5-17 (immunoglobulin heavy variable 5-17) activates the immune response, defense response, and phagocytosis and has been reported to be a potential prognostic biomarker for acute myeloid leukemia [96]. Myo18a regulates intracellular transport processes and activates innate immune receptors on macrophages, B cell development, homeostasis, and antibody-mediated immunity [97,98,99]. Gcdh (Glutaryl-CoA dehydrogenase, EC 1.3.8.6) enzyme is a flavoprotein involved in the metabolism of glutaryl-CoA that mainly occurs in mitochondria [100]. It is known that Gcdh degrades glutaryl-CoA, thereby reducing the lysine glutarylation (Kglu) level of the protein [101]. Kglu plays a crucial role in the mitochondrial and metabolic processes, regulating various cellular functions such as amino acid metabolism, fatty acid metabolism, cellular respiration, and protein function [102,103,104,105]. The abnormal lysine acetylation affects multiple cellular processes and is pathogenic in diverse conditions such as metabolic syndrome, cardiac failure, and cancer [106,107,108]. Recently, Zhang et al. [109] demonstrated that Gcdh is overexpressed in cervical cancer. Its expression was correlated with tumor-infiltrating macrophage and immunity. Therefore, the upregulation of Ighv5-17, Myo18a, and Gcdh suggests that svCRiSP might play an important role in the early development of immune responses in snakebite.

Pdcl3 (Phosducin Like 3) is a chaperone protein that plays an important role regulating the stability and function of vascular endothelial growth factor receptor (VEGFR-2) [110]. VEGFR-2 is mainly distributed in vascular endothelial cells and mediates signaling transduction, biological responses, and pathological processes in angiogenesis through PLCγ-PKC-MAPK, PLCγ-PKC-eNOS-NO, TSAd-Src-PI3K-Akt, SHB-FAK-paxillin, SHB-PI3K-Akt, and NCK-p38-MAPKAPK2/3 pathways [111,112]. Furthermore, cellular signaling mediated by VEGFR-2 activates downstream signaling pathways and ultimately affects endothelial cell survival, proliferation, cell migration, and vascular permeability [113]. This suggests that Pdcl3-mediated VEGFR-2 signaling may be one important molecular mechanism of increased vascular permeability induced by svCRiSPs. Notably, the plasma EV proteome did not detect the top downregulated proteins in common in both P-vEV and P-CRiSP-EV, although only 11 proteins were identified.

Due to the diverse array of toxin components in crude venom and their complex biochemical interactions, various endogenous signaling systems can be activated to induce cell injury and systemic effects [114]. This intricate mixture of toxins can lead to results beyond what would be expected from the direct impact of a whole venom dose or individual toxins, making it challenging to observe downregulated proteins in common between P-vEV and P-CRiSP-EV.

While the results of this study align with previously reported information on the biological functions and cellular signaling of svCRiSPs [11,12], further investigations are needed to validate these findings. Moreover, delving into the biological functions of other protein families and venoms could provide valuable insights into their roles in the wide range of clinical effects observed in snake venoms. This exploration can also provide a broader perspective on possible avenues for future research and the importance of investigating other protein families. Our study’s value lies in the potential to deepen our understanding of the cellular and molecular mechanisms of svCRiSPs and their role in the intricate pathophysiological responses triggered by snake envenoming.

## 4. Conclusions

In the present study, we showed the contribution of svCRiSP in the pathophysiology of snakebites and suggested some early envenomation markers, such as Exoc8 and Apoa2. However, to be able to precisely indicate whether any specific biomarkers that exist are snakebite specific would require extensive secondary validation and additional assays beyond the scope of the experiments presented. Exploring EVs has the potential to advance our comprehension of the molecular-level pathogenesis of snakebites. By enabling early detection and assessment of local effects, it is possible to mitigate the complications arising from severe systemic effects, long-term disabilities, and fatalities and facilitate targeted therapeutic strategies.

## 5. Materials and Methods

### 5.1. C. s. scutulatus Venom and Its Purified CRiSP

Lyophilized crude venom from *C. s. scutulatus* (Mohave rattlesnake, Type A) was obtained from an individual adult snake housed in the National Natural Toxins Research Center (NNTRC) serpentarium, located at Texas A&M University-Kingsville, Kingsville, TX. Venom extraction was performed by placing disposable plastic cups covered with parafilm in close proximity to the snakes. The extracted venom was subsequently centrifuged at 10,000× *g* at 4 °C for 5 min using a Beckman Coulter Avanti 30 Centrifuge. The venom was then filtered through a 0.45 µm MillexHV syringe filter unit (Millipore, Billerica, MA, USA), followed by lyophilization. Finally, the lyophilized venoms were stored at −80 °C until further use. Css-CRiSP was purified as described by Suntravat et al. [12].

### 5.2. Animals and Experimental Envenomation

Groups of ten BALB/c mice (male and female, 18–21 g body weight) were used. The mice were housed in temperature-controlled rooms at 23–25 °C on a 12 h light/dark cycle and received water and food ad libitum until used. The mice were divided into three groups: (1) crude venom; (2) purified Css-CRiSP; and (3) normal saline injections. Animal protocols were approved by the Institutional Animal Care and Use Committee according to protocols ratified by the NNTRC (Viper Resource Center at Texas A&M University-Kingsville, IACUC #: 2021-11-29/1474).

Mice (*n* = 10/group) were intraperitoneally injected with 100 µL of sublethal dose of *C. s. scutulatus* crude venom (68.5 μg/mouse), Css-CRiSP (2 µg/mouse), and 0.85% NaCl (*w*/*v*) (negative control). The usage dose of Css-CRiSP was calculated by the relative proteomic abundances of Css-CRiSP in the total venom (~3%) based on the proteomic analysis of this venom [12]. After 30 min, mice were euthanized by CO_2_ inhalation. Blood was immediately collected from the heart into 0.1 M EDTA tubes. A midline incision was performed in the abdominal cavity, and peritoneal exudate was drained. The exudate was then incubated with polymyxin B (15 μg/mL), a well-characterized pharmacologic LPS antagonist. The blood and peritoneal exudate were then centrifuged and then stored at −80 °C for subsequent determination of EV content using the EVTrap method as previously described [35].

### 5.3. Purification of EVs

Approximately 10 μL of peritoneal exudate and plasma was separately pooled from each mouse. Plasma and peritoneal exudate EVs were purified using the EVTrap method, as described previously [56]. Briefly, the plasma and exudate samples were centrifuged at 2500× *g* for 10 min at room temperature to remove platelets and apoptotic bodies and carefully collect the supernatant. Next, the supernatants were diluted 20 times in the diluent buffer, which was provided with the EVtrap beads as part of the kit, and the EVtrap beads added in a 1:2 *v*/*v* ratio. The samples were then incubated with end-over-end rotation for 30 min, following the instructions provided by the manufacturer. After removing the supernatant using a magnetic separator rack, the beads were washed with PBS, and the plasma EVs (pEV) and exudate EVs (E-EVs) were eluted by incubating with 200 mM triethylamine (TEA, Milli-pore-Sigma, MO, USA) for 10 min. Finally, the samples were completely dried using a vacuum centrifuge and stored at −80 °C.

### 5.4. LC-MS/MS Analysis

LC-MS/MS analysis was performed according to Willard et al. [35]. Briefly, for each dried peptide sample, 1 µg was dissolved in 10.5 μL of a solution containing 0.05% trifluoroacetic acid in 3% (vol/vol) acetonitrile. Subsequently, 10 μL of each sample was injected into an Ultimate 3000 nano UHPLC system (Thermo Fisher Scientific, MA, USA). Peptides were loaded on a 2 cm Acclaim PepMap trap column and then separated on a heated 50 cm column packed with ReproSil Saphir 1.8 μm C18 beads (Dr. Maisch GmbH, Ammerbuch, Germany). The mobile phase buffer comprised 0.1% formic acid in ultrapure water (buffer A), while the eluting buffer consisted of 0.1% formic acid in 80% (*v*/*v*) acetonitrile (buffer B). The separation process involved a linear 60 min gradient of 6% to 30% buffer B at 300 nL/min flow rate. The UHPLC system was coupled online with a Q Exactive HF-X mass spectrometer (Thermo Fisher Scientific). The mass spectrometer operated in the data-dependent mode, starting with a full-scan MS analysis from *m*/*z* 375 to 1500, employing a resolution of 60,000. This was followed by MS/MS analysis of the 15 most intense ions, utilizing a resolution of 30,000. The normalized collision energy was set at 28%, and the automatic gain control target (AGC) was set to 2 × 10^4^ with a maximum injection time of 200 ms. Additionally, a 60-s exclusion window was implemented.

The raw LC-MS/MS data were searched directly against all the *Mus musculus* proteins available in Uniprot (downloaded December 2022) without redundant entries. Byonic (Protein Metrics, Cupertino, CA, USA) and SEQUEST search engines were utilized within Proteome Discoverer 2.3 software (Thermo Fisher Scientific). The MS1 precursor mass tolerance was set at 10 ppm, and the MS2 tolerance was set at 20 ppm. Search criteria involved a static modification of carbamidomethylation on cysteines (+57.0214 Da), variable modifications of oxidation on methionine residues (+15.9949 Da), and acetylation at the N-terminus of proteins (+42.011 Da). The search was conducted using complete trypsin/P digestion, allowing a maximum of two missed cleavages on the analyzed peptides from the sequence database. Protein and peptide identifications were subjected to a 1% false-discovery rate (FDR). All identified proteins and peptides were grouped, eliminating any redundant entries. Unique peptides and unique master proteins were reported.

### 5.5. Data Acquisition, Quantification, and Bioinformatic Analysis

Label-free quantitation node of Precursor Ions Quantifier by way of the Proteome Discoverer v2.3 (Thermo Fisher Scientific) was used to quantify the data. Statistical analysis of total peptides and protein abundance was conducted first by normalization using the total peptide normalization zone in the proteome discoverer. Then, data processing proteomic quantification of both plasma and exudate sample intensities of peptides were extracted with initial precursor mass tolerance set at 10 ppm and PSM confidence FDR rate 1%. To ensure consistent comparisons, the abundance levels of all peptides and proteins were normalized using the total peptide amount normalization node in Proteome Discoverer.

To compare the lists of protein groups identified in the P-EV and E-EV datasets, Venn diagrams were created using the VENNY software version 2.1 (assessed April 2023). Furthermore, the identified protein groups were cross-referenced with the EV markers found in the ExoCarta Top100 database (http://exocarta.org/exosome_markers_new, accessed on 20 April 2023) and the Vesiclepedia Top100 database (http://microvesicles.org/extracellular_vesicle_markers, accessed on 20 April 2023) to assess the efficacy of the EV purification method. Fold change (FC) was calculated by subtracting the average log2 values between proteins identified in treatment groups vs. proteins identified in the negative control group. The comparison was performed in four pairwise groups, i.e., E-vEV vs. E-cEV, E-CRiSP-EV vs. E-cEV, P-vEV vs. P-cEV, P-CRiSP-EV vs. P-cEV, using Student’s *t*-test for two-tailed unpaired data. A *p*-value of 0.05 with a log2 FC > 1 and <−1 was set as the cutoff for determining whether a protein was differentially expressed. Heatmaps, Volcano plots, and *p*-values were generated using Perseus software (Version 2.0.6.0) [115]. Volcano plots were drawn in RStudio (Version 2022.07.2). The proteins were subjected to gene ontology (GO) analysis using the Database for Annotation, Visualization, and Integrated Discovery (DAVID, version 2021). The GO terms were categorized into three groups: biological process (BP), cellular component (CC), and molecular function (MF). The upregulated and downregulated proteins were analyzed separately, and a significance level of *p* < 0.05 was used to determine statistically significant differences. The protein–protein interaction (PPI) network was generated using the significantly expressed proteins and the Search Tool for the Retrieval of Interacting Genes/Proteins (STRING) version 11.5 database (http://string-db.org/, accessed on 18 April 2023). The species “*Mus musculus*” was chosen for constructing the PPI network, and a full STRING network was selected. The network required a high confidence score threshold of 0.400 and an FDR stringency of 5% (medium).

## Figures and Tables

**Figure 1 toxins-15-00434-f001:**
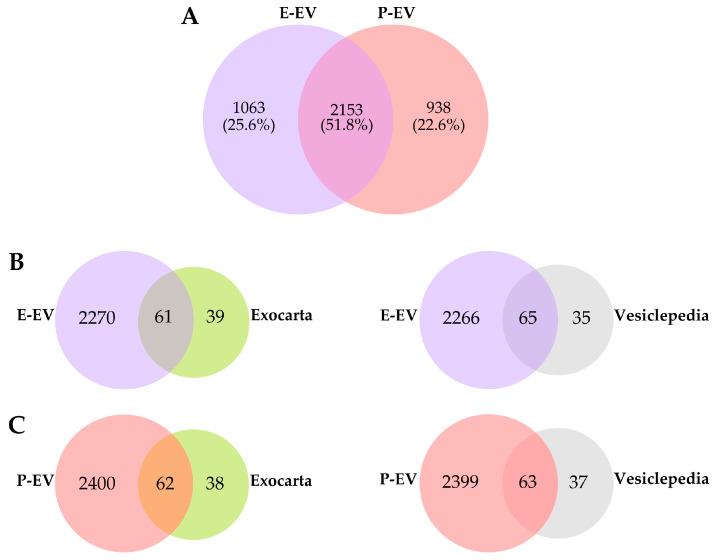
Venn diagrams of proteins identified in EVs. (**A**) Comparison of proteins identified in EVs from peritoneal exudate (E-EV) and plasma (P-EV). Venn diagrams showing comparison of E-EV (**B**) and P-EV (**C**) with the top 100 proteins most often identified in the EVs/exosomes from ExoCarta and Vesiclepedia databases. (*n* = 10/group).

**Figure 2 toxins-15-00434-f002:**
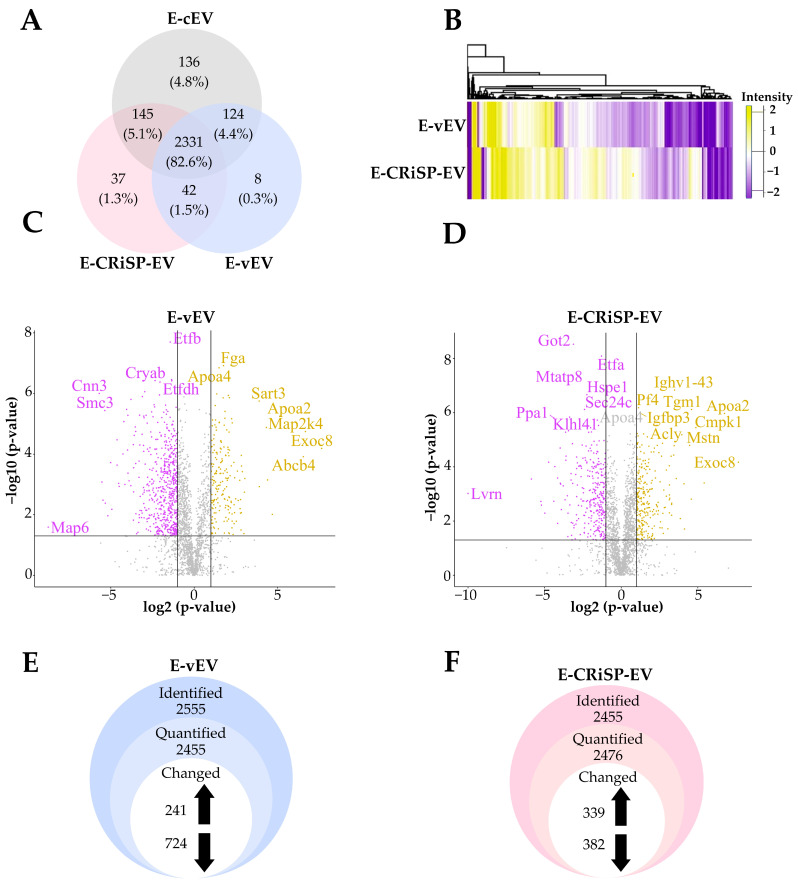
Proteomic analysis of E-vEV, E-CRiSP-EV, and E-cEV. (**A**) The Venn diagram shows the overlap of proteins identified in P-vEV, P-CRiSP-EV, and P-cEV, present in the technical triplicates. (**B**) Heatmap representing the log2 FC of the 2331 commonly expressed proteins in E-vEV and E-CRiSP-EV after unsupervised hierarchical clustering. Downregulated and upregulated proteins are colored in purple and yellow, respectively. Proteins without significant changes are shown in white. Volcano plots show upregulated and downregulated proteins in E-vEV (**C**) and E-CRiSP-EV (**D**) compared to control (E-cEV). The yellow and purple circles indicate significant upregulated (log2 FC > 1) and downregulated (log2 FC < −1) proteins, respectively. The black line indicates the limit of statistical significance (*p* < 0.05). The Y-axis represents the −log10 *p*-value. The X-axis represents the log2 FC of the relative protein abundances in the treatment groups versus the control. Changes identified from label-free protein quantification in E-vEV (**E**) and E-CRiSP-EV (**F**). (*n* = 10/group).

**Figure 3 toxins-15-00434-f003:**
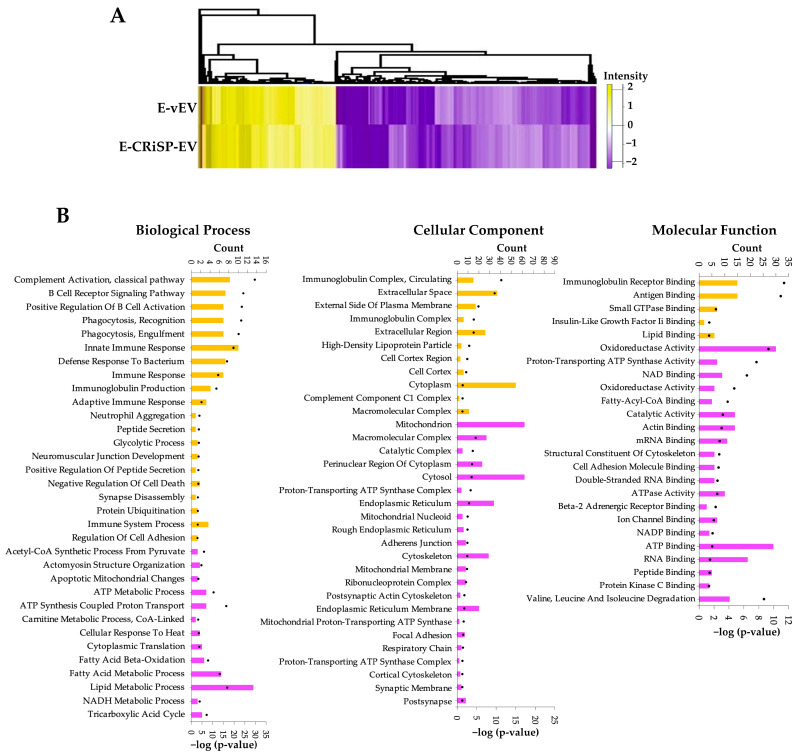
The profiles of overlapping protein with significant changes in both E-vEV and E-CRiSP-EV. (**A**) The heatmap represents the log2 FC of the significantly regulated proteins in E-vEV and E-CRiSP-EV. (**B**) GO analysis of selected proteins enriched in both EV groups. The upregulated (yellow) and downregulated (purple) terms in three main categories, including biological process, cellular components, and molecular functions are shown and sorted by *p*-value (*n* = 10/group).

**Figure 4 toxins-15-00434-f004:**
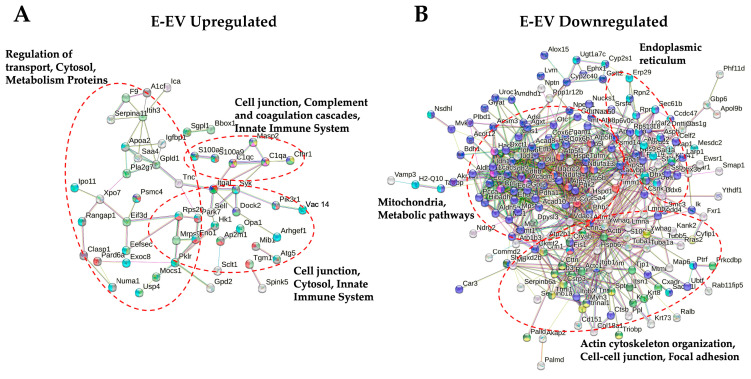
Protein–protein interaction networks of proteins significantly altered in E-EV and E-CRiSP-EV using the STRING database (v. 11.5). Direct protein–protein interactions were selected with 0.4 confidence. (**A**) Upregulated protein interaction network and (**B**) downregulated protein interaction network (*n* = 10/group).

**Figure 5 toxins-15-00434-f005:**
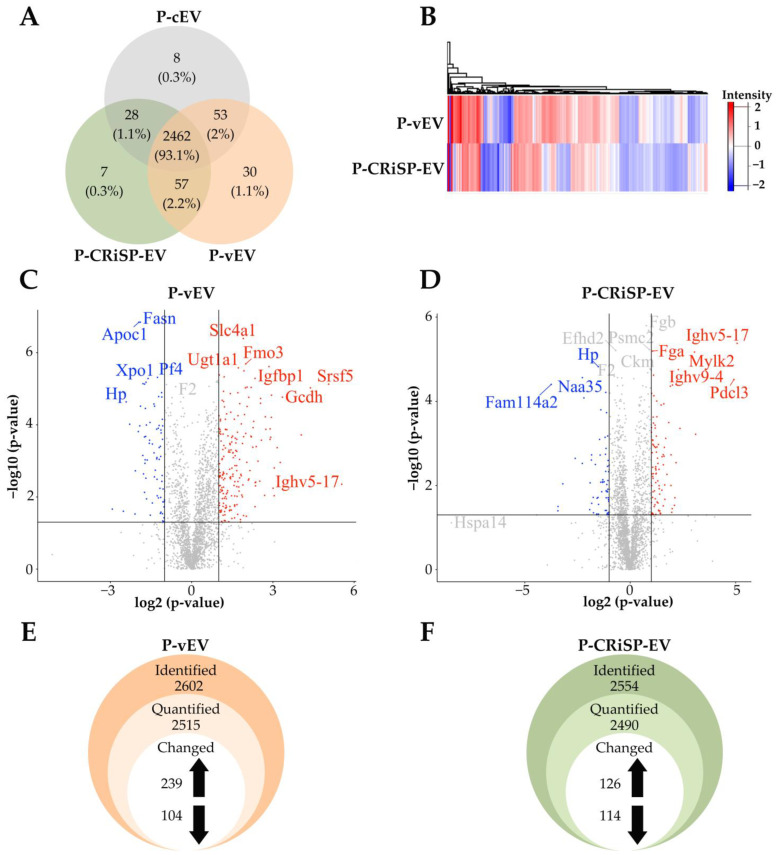
Proteomic analysis of P-vEV, P-CRiSP-EV, and P-cEV. (**A**) The Venn diagram shows the overlap of proteins identified in P-vEV, P-CRiSP-EV, and P-cEV, present in the technical triplicates. (**B**) Heatmap representing the log2 FC of the 2462 commonly expressed proteins in P-vEV and P-CRiSP-EV after unsupervised hierarchical clustering. Downregulated and upregulated proteins are colored in blue and red, respectively. Proteins without significant changes are shown in white. Volcano plots show upregulated and downregulated proteins in P-vEV (**C**) and P-CRiSP-EV (**D**). The red and blue circles indicate significant upregulated (log2 FC > 1) and downregulated (log2 FC < −1) proteins, respectively. The black line indicates the limit of statistical significance (*p* < 0.05). The Y-axis represents the −log10 *p*-value. The X-axis represents the log2 FC of the relative protein abundances in the treatment groups versus control. Changes identified from label-free quantitation in P-vEV (**E**) and P-CRiSP-EV (**F**). (*n* = 10/group).

**Figure 6 toxins-15-00434-f006:**
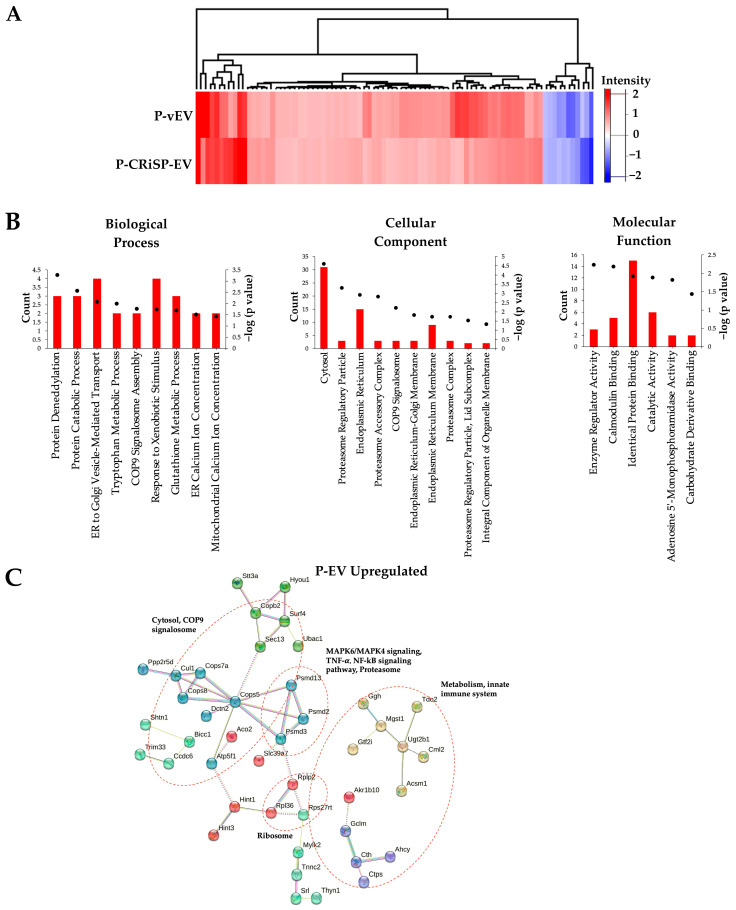
Proteomic analysis of significantly altered proteins in both P-vEV and P-CRiSP-EV. (**A**) Heatmap representing the log2 FC of the commonly identified proteins in both P-vEV and P-CRiSP-EV. The color scale shown in the map illustrates the relative protein expression: red represents upregulated proteins, and blue represents downregulated proteins. (**B**) GO enrichment analysis using DAVID for upregulated proteins in P-vEV and P-CRiSP-EV. The red bars represent the number of proteins, and the black dots indicate *p* < 0.05. The enriched GO terms in the biological process, cellular component, and molecular function categories are represented. (**C**) The protein–protein network analysis using the STRING database for the upregulated proteins in P-vEV and P-CRiSP-EV. (*n* = 10/group).

**Figure 7 toxins-15-00434-f007:**
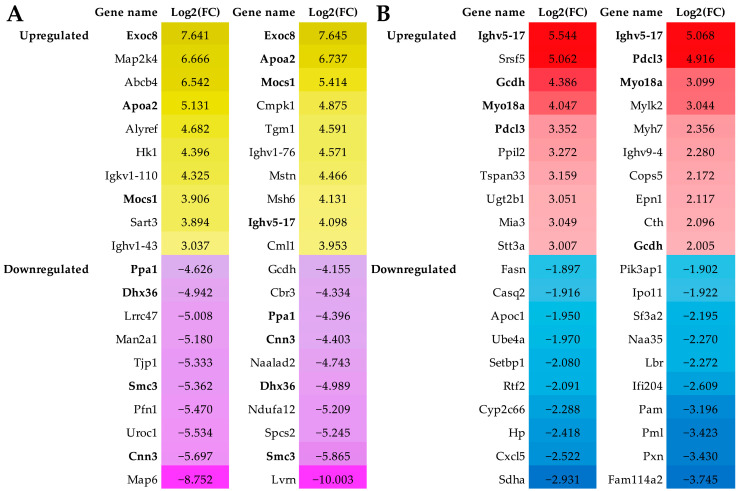
The top 10 upregulated and downregulated proteins and their respective log2 FC (*p* < 0.05) of vEV and CRiSP-EV from exudate (**A**) and plasma (**B**). The color scales indicate the relative protein expression compared to the control group: yellow and red represent upregulated proteins in the exudate and plasma, respectively; purple and blue represent downregulated proteins in the exudate and plasma. Proteins in both vEV and CRiSP-EV top ten lists are represented in bold. (*n* = 10/group).

## Data Availability

The datasets utilized and analyzed in the present study can be obtained from the corresponding author upon reasonable request.

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
