# Peer review of "Proteomic Profiling of Extracellular Vesicles Isolated from Plasma and Peritoneal Exudate in Mice Induced by Crotalus scutulatus scutulatus Crude Venom and Its Purified Cysteine-Rich Secretory Protein (Css-CRiSP)"

_toxins, 2023, doi:10.3390/toxins15070434_

Round 1

Reviewer 1 Report

1.  Viper venoms, as correctly stated in the introduction, contain several proteins belonging to roughly ten protein families. It would be interesting to test other proteins vs. venom in the same way as it has been done for CRiSP. Then it would be possible to identify all the proteins involved in the same process. At present it would only be appropriate to discuss/mention this point in the discussion or at the end of introduction.

There are a couple of instances of  exactly the same sentence(s) in the present manuscript as the cited literature, although the sentences were not cited. It is possible that the authors "constructed" the same sentence by themselves, so I did not mention detection of plagiarism, but i strongly suggest that the authors make a thorough plagiarism check and add appropriate references or change the text to avoid any doubt of a copy/paste possibility (an example: Apolipoprotein A-II (apoA-II) is the second most abundant apolipoprotein in high-density lipoprotein (HDL) particles, playing an important role in lipid metabolism. pThe same sentence appears in the article published by Florea G in Biomedicines in 2022)

Otherwise the manuscript is relatively easy to read. Supplementary material is abundant, but appropriate for the type of publication.

Author Response

Point 1 Viper venoms, as correctly stated in the introduction, contain several proteins belonging to roughly ten protein families. It would be interesting to test other proteins vs. venom in the same way as it has been done for CRiSP. Then it would be possible to identify all the proteins involved in the same process. At present it would only be appropriate to discuss/mention this point in the discussion or at the end of introduction.

Response 1: We appreciate your recognition of the importance of exploring additional proteins in viper venoms beyond the ones discussed in our study, particularly in identifying all the proteins involved in the same process. Exploring the biological functions of protein families, much like this investigation on svCRiSP, could provide valuable insights into their contributions to the diverse clinical effects of snake venoms. This point was mentioned in the discussion section.

Point 2 There are a couple of instances of exactly the same sentence(s) in the present manuscript as the cited literature, although the sentences were not cited. It is possible that the authors "constructed" the same sentence by themselves, so I did not mention detection of plagiarism, but i strongly suggest that the authors make a thorough plagiarism check and add appropriate references or change the text to avoid any doubt of a copy/paste possibility (an example: Apolipoprotein A-II (apoA-II) is the second most abundant apolipoprotein in high-density lipoprotein (HDL) particles, playing an important role in lipid metabolism. pThe same sentence appears in the article published by Florea G in Biomedicines in 2022)

Otherwise the manuscript is relatively easy to read. Supplementary material is abundant, but appropriate for the type of publication.

Response 2: Thank you for thoroughly reviewing our manuscript and bringing up the issue regarding instances of similar sentences without proper citation. Upon further examination and based on your example, we acknowledge a sentence in our manuscript that closely resembles a sentence published by Florea G in Biomedicines in 2022. We apologize for this oversight. In response to your suggestion, we have conducted a thorough plagiarism check using reliable software, and apart from the small phrases detected within some sentences, no other significant instances of plagiarism were found.

Reviewer 2 Report

In the manuscript entitled “Proteomic profiling of extracellular vesicles isolated from plasma and peritoneal exudate in mice induced by Crotalus scutulatus scutulatus crude venom and its purified cysteine-rich secretory protein (Css-CRiSP)”, the authors present a detailed proteomic characterization of the contents of extracellular vesicles from plasma and peritoneal exudate of envenomated mice and compare the effects of whole C. s. scutulatus venom with those of a purified CRiSP from the same venom and a non-envenomated control. I believe the research is properly designed and the results are relevant and correctly presented.

Therefore, I only have some minor comments for the authors:

 Line 331. The authors mention that the identification of 60 of the top 100 EV proteins in public databases indicates the successful isolation of EVs. Is there a non-arbitrary reason for this? What would be considered a non-successful isolation?

Line 347. Please specify which acute effects induced by CRiSP proteins have been documented. This sentence seems to imply that all envenomation can be attributed to them, and that is not the case.

Line 396. The sentence reads “…further investigation of Apoa2 as a potential systemic inflammation in snakebite is suggested.”. Please clarify if this should read “…systemic inflammation marker…” or rewrite for clarity.

Line 453. Do the authors have any theories regarding why the top downregulated proteins in common between P-vEV and P-CRiSP-EV were not detected?

Line 484. Please specify the injection route.

Line 501. Please provide the components of the diluent buffer.

Figures 2 and 5. I think the figures´ color scheme could be improved. It is a bit confusing because the same colors mean different things throughout the figures. Also, colors are inverted between A and E/F, e.g., E-CRiSP-EV purple in A and yellow in F.

 Figure 3 and 4. Fonts are very small and hard to read. If possible, please increase the front size.  

Figure 7. The legend mentions A and B, but there are no A and B letters in the figure.

Very minor language edits:

Line 8. It is not clear what is meant by “common sequence”.

Line 17. The phrase “… after 30 min post-injection…” seems redundant

Line 37. Seems like the word “heterogenetic” is not correctly used here. I believe it should be heterogeneous, or similar.

Line 38. The 2 in phospholipases A2 should be subindex.

Line 477. Should read BALB/c instead of BALB.

Line 479. Ad libitum should be italicized.

Author Response

In the manuscript entitled “Proteomic profiling of extracellular vesicles isolated from plasma and peritoneal exudate in mice induced by Crotalus scutulatus scutulatus crude venom and its purified cysteine-rich secretory protein (Css-CRiSP)”, the authors present a detailed proteomic characterization of the contents of extracellular vesicles from plasma and peritoneal exudate of envenomated mice and compare the effects of whole C. s. scutulatus venom with those of a purified CRiSP from the same venom and a non-envenomated control. I believe the research is properly designed and the results are relevant and correctly presented.

Response 1: We thank the reviewer for the valuable comments.

Therefore, I only have some minor comments for the authors:

Point 2 Line 331. The authors mention that the identification of 60 of the top 100 EV proteins in public databases indicates the successful isolation of EVs. Is there a non-arbitrary reason for this? What would be considered a non-successful isolation?

Response 2: Several reports have emphasized the significant value of utilizing Vesiclepedia and Exocarta as invaluable resources for identifying molecular signatures previously characterized within specific EV populations (Kalra et al., 2012; Keerthikumar et al., 2016). Previous studies have demonstrated a wide range (23% - 95%) of identified EV markers in these top 100 EV databases, reflecting the efficacy of the EV purification method. This variation can be attributed to the diverse EV isolation methods employed and the various sample types, such as exosomes derived from primary cells, cell cultures, tissue cultures, and different biological fluids (Reymond et al., 2023; Alameldin et al., 2021; Heath et al., 2018; Konoshenko et al., 2018; Yang et al., 2017). Therefore, the results obtained in our study provide substantial evidence for the reliability of our EV isolation method.

References

Kalra H, Simpson RJ, Ji H, Aikawa E, Altevogt P, Askenase P, Bond VC, Borràs FE, Breakefield X, Budnik V, Buzas E, Camussi G, Clayton A, Cocucci E, Falcon-Perez JM, Gabrielsson S, Gho YS, Gupta D, Harsha HC, Hendrix A, Hill AF, Inal JM, Jenster G, Krämer-Albers E-M, Lim SK, Llorente A, Lötvall J, Marcilla A, Mincheva-Nilsson L, Nazarenko I, Nieuwland R, Hoen ENMN-’t, Pandey A, Patel T, Piper MG, Pluchino S, Prasad TSK, Rajendran L, Raposo G, Record M, Reid GE, Sánchez-Madrid F, Schiffelers RM, Siljander P, Stensballe A, Stoorvogel W, Taylor D, Thery C, Valadi H, Balkom BWM van, Vázquez J, Vidal M, Wauben MHM, Yáñez-Mó M, Zoeller M, Mathivanan S. Vesiclepedia: a compendium for extracellular vesicles with continuous community annotation. PLOS Biolog. 2012;10 e1001450.

Keerthikumar S, Chisanga D, Ariyaratne D, Saffar HA, Anand S, Zhao K, Samuel M, Pathan M, Jois M, Chilamkurti N, Gangoda L, Mathivanan S. ExoCarta: a web-based compendium of exosomal cargo. J Mol Biol. 2016;428:688–92.

Reymond S, Gruaz L, Sanchez JC. Depletion of abundant plasma proteins for extracellular vesicle proteome characterization: benefits and pitfalls. Anal Bioanal Chem. 2023 Apr 18. doi: 10.1007/s00216-023-04684-w. Epub ahead of print. PMID: 37069444.

Alameldin S, Costina V, Abdel-Baset HA, Nitschke K, Nuhn P, Neumaier M, Hedtke M. Coupling size exclusion chromatography to ultracentrifugation improves detection of exosomal proteins from human plasma by LC-MS. Pract Lab Med. 2021 Jun 23;26:e00241. doi: 10.1016/j.plabm.2021.e00241. PMID: 34258353; PMCID: PMC8254000.

Heath N, Grant L, De Oliveira TM, Rowlinson R, Osteikoetxea X, Dekker N, Overman R. Rapid isolation and enrichment of extracellular vesicle preparations using anion exchange chromatography. Sci Rep. 2018 Apr 10;8(1):5730. doi: 10.1038/s41598-018-24163-y. PMID: 29636530; PMCID: PMC5893571.

Konoshenko MY, Lekchnov EA, Vlassov AV, Laktionov PP. Isolation of Extracellular Vesicles: General Methodologies and Latest Trends. Biomed Res Int. 2018 Jan 30;2018:8545347. doi: 10.1155/2018/8545347. PMID: 29662902; PMCID: PMC5831698.

Yang C, Guo WB, Zhang WS, Bian J, Yang JK, Zhou QZ, Chen MK, Peng W, Qi T, Wang CY, Liu CD. Comprehensive proteomics analysis of exosomes derived from human seminal plasma. Andrology. 2017 Sep;5(5):1007-1015. doi: 10.1111/andr.12412. PMID: 28914500; PMCID: PMC5639412.

Point 3 Line 347. Please specify which acute effects induced by CRiSP proteins have been documented. This sentence seems to imply that all envenomation can be attributed to them, and that is not the case.

Response 3: We agree with the reviewer to specify the acute effects induced by CRiSPs. We have revised the statement as follows: “Viperine CRiSPs have been previously shown to induce vascular permeability, activate key components of cellular signaling pathways, and promote inflammatory responses [11-13].”

Point 4 Line 396. The sentence reads “…further investigation of Apoa2 as a potential systemic inflammation in snakebite is suggested.”. Please clarify if this should read “…systemic inflammation marker…” or rewrite for clarity.

Response 4: Thank you for pointing this out. Corrected.

Point 5 Line 453. Do the authors have any theories regarding why the top downregulated proteins in common between P-vEV and P-CRiSP-EV were not detected?

Response 5: Due to the diverse array of toxin components in crude venom and their complex biochemical interactions, various endogenous signaling systems can be activated to induce cell injury and systemic effects (Bickler, 2020). This intricate mixture of toxins can lead to results beyond what would be expected from the direct impact of a whole venom dose or individual toxins, making it challenging to observe downregulated proteins in common between P-vEV and P-CRiSP-EV. We added this statement in the discussion section.

Reference

Bickler PE. Amplification of Snake Venom Toxicity by Endogenous Signaling Pathways. Toxins (Basel). 2020 Jan 22;12(2):68. doi: 10.3390/toxins12020068. PMID: 31979014; PMCID: PMC7076764.

Point 6 Line 484. Please specify the injection route.

Response 6: Thank you for pointing this out. Corrected.

Point 7 Line 501. Please provide the components of the diluent buffer.

Response 7:  The diluent buffer was provided with the EVtrap beads as part of the kit. We added this sentence in the method section.

Point 8 Figures 2 and 5. I think the figures´ color scheme could be improved. It is a bit confusing because the same colors mean different things throughout the figures. Also, colors are inverted between A and E/F, e.g., E-CRiSP-EV purple in A and yellow in F.

 Figure 3 and 4. Fonts are very small and hard to read. If possible, please increase the front size.  

Figure 7. The legend mentions A and B, but there are no A and B letters in the figure.

Response 8: Thank you for pointing this out. Corrected.

Comments on the Quality of English Language

Very minor language edits:

Point 9 Line 8. It is not clear what is meant by “common sequence”.

Response 9: Thank you for pointing this out. Corrected.

Point 10 Line 17. The phrase “… after 30 min post-injection…” seems redundant

Response 10: Thank you for pointing this out. Corrected.

Point 11 Line 37. Seems like the word “heterogenetic” is not correctly used here. I believe it should be heterogeneous, or similar.

Response 11: Thank you for pointing this out. Corrected.

Point 12 Line 38. The 2 in phospholipases A2 should be subindex.

Response 12: Thank you for pointing this out. Corrected.

Point 13 Line 477. Should read BALB/c instead of BALB.

Response 13: Thank you for pointing this out. Corrected.

Point 14 Line 479. Ad libitum should be italicized.

Response 14: Thank you for pointing this out. Corrected

Reviewer 3 Report

The authors present an investigation designed to identify proteins in peritoneal exudate and plasma extracellular vesicles (EVs) isolated from BALB/c mice after 30 min post-injection of Crotalus scutulatus scutulatus venom and its purified cysteine-rich secretory proteins [CRiSP (Css-CRiSP)]. The rationale for the investigation was that venomous snake bite causes a variety of local and systemic symptoms that are based on changes in vascular permeability, and EVs play a role in these pathophysiological phenomena.

Results

In general, the figures are somewhat difficult to read, with figure 2, panel B being essentially uninterpretable with the density of information presented with unreadable font. The remaining panels can be interpreted, but the fonts are still quite small, and yellow coloration decreases the ability to read the information provided. Figure 3A is uninterpretable, and 3B could be improved with increased size to increase the font. Figures 4 & 5 can be improved but are greatly improved compared to the other figures. Without belaboring the point further, it would be great to improve the figures to assist the readership in understanding the message.

While provided in the Methods section, it would be helpful to indicate the number of animals used in the figure legends or text in results.

Discussion & Conclusions

The authors fairly frame their results, and the cataloguing of proteins/protein families in EVs obtained from the peritoneal cavity and plasma of  envenomed mice is very interesting.

However, I have only one real concern. While the dose of venom was sublethal, it is not indicated what type of Mojave rattlesnake venom was used – the type A, wherein the mode of death is via paralysis with Mojave toxin that contains phospholipase A2 activity, or the type B that is devoid of Mojave toxin but instead is resplendent with proteolytic enzymes that cause coagulopathy. Why this is important is that if type A was used, it is possible the animals experienced hypoxia during envenomation, which would be a “two-hit” model of injury as opposed to the effects of Css-CRiSP mediated effects that would be presumed to be devoid of hypoxia mediated effects. Thus, it would be helpful if the authors could clarify if the effects observed are secondary to the effects of the crude venom without/with hypoxia versus the effects of Css-CRiSP.

Author Response

Point 1 Results

In general, the figures are somewhat difficult to read, with figure 2, panel B being essentially uninterpretable with the density of information presented with unreadable font. The remaining panels can be interpreted, but the fonts are still quite small, and yellow coloration decreases the ability to read the information provided. Figure 3A is uninterpretable, and 3B could be improved with increased size to increase the font. Figures 4 & 5 can be improved but are greatly improved compared to the other figures. Without belaboring the point further, it would be great to improve the figures to assist the readership in understanding the message.

Response 1: We appreciate your thorough evaluation and concerns regarding the figures. We made improvements to address the font size and readability concerns. We have increased the font size across all figures, ensuring better legibility and making the information more accessible to readers.

Point 2 While provided in the Methods section, it would be helpful to indicate the number of animals used in the figure legends or text in results.

Response 2: Thank you for your feedback regarding the indication of the number of animals used in our study. We considered your suggestion of including the number of animals used in the figure legends for every figure.

Point 3 Discussion & Conclusions

The authors fairly frame their results, and the cataloguing of proteins/protein families in EVs obtained from the peritoneal cavity and plasma of envenomed mice is very interesting.

However, I have only one real concern. While the dose of venom was sublethal, it is not indicated what type of Mojave rattlesnake venom was used – the type A, wherein the mode of death is via paralysis with Mojave toxin that contains phospholipase A2 activity, or the type B that is devoid of Mojave toxin but instead is resplendent with proteolytic enzymes that cause coagulopathy. Why this is important is that if type A was used, it is possible the animals experienced hypoxia during envenomation, which would be a “two-hit” model of injury as opposed to the effects of Css-CRiSP mediated effects that would be presumed to be devoid of hypoxia mediated effects. Thus, it would be helpful if the authors could clarify if the effects observed are secondary to the effects of the crude venom without/with hypoxia versus the effects of Css-CRiSP.

Response 3: We appreciate the reviewer’s comments. As mentioned, the different biochemical properties exhibited by Mojave rattlesnake venoms will cause phenotypical variations in the snakebite envenomings, producing a diverse range of clinical manifestations, where those caused by the type A group will lack the common local tissue effects observed in most crotalid envenomings, while the ones caused by the highly proteolytic type B group, will display hemorrhagic manifestations and classical local tissue damage. 

The aim of this study was evaluating the early effects of svCRiSP and crude venom using proteomic analysis of EVs purified from different biofluids collected from our experimental model, thus providing more insights about the early events in the pathophysiology of snakebites and the relevant role a minor protein family (svCRiSP) may exert. Our experiment was conducted for a short period using a sub-lethal dose of type A Mojave rattlesnake venom. We specified the type of Mojave rattlesnake venom used in this study.

We did not observe distress in the animals or any evident symptom suggesting neurological alterations. In addition, the alterations and possible hypoxia that might develop in the experimental group could be a consequence of the action of other snake venom toxins, where Css-CRiSP may be involved by regulating some of the early events seen in the snakebite. Regardless, further experiments would be required to show if these effects are caused solely by Css-CRiSP or in a synergistic manner with other toxins families found in the Mojave rattlesnake venom.

Reviewer 4 Report

The authors identify proteins in peritoneal exudate and plasma EVs isolated from BALB/c mice after 30 min post-injection of Crotalus scutulatus scutulatus venom and its purified CRiSP (Css-CRiSP). Moreover this study investigated the proteome expression profile of EVs isolated from biofluid samples using a label-free LC-MS/MS method. The discussion and conclusion should be improved. Thus I send my considerations and some questions need to be clarified.

Results

Line 128

...specific to E-cEV (Figure 2A). We found that 2,455 proteins were quantified in E-vEV (Figure 2E), while 2,476 proteins were quantified in E- 129

the authors could present the results in the same sequence that they present in the figure, making the reading easier

Figure 2.

the graphs presented in purple and yellow could be showed in different colors from those used to express the  upregulated and downregulated proteins to avoid confusion in data analysis. The same pattern should be applied in figure 5 and 6 instead of using blue and red colors to represent upregulated and downregulated proteins

2.4.

Line 262:  The authors should explain with more refinement about the statement in relation to downregulation of exudate expression with upregulation of plasma protein expression.

3. Discussion

he discussion is very extensive. I suggest that the authors synthesize this topic and contest and/or deepen the discussion of some points such as those presented in lines 309-31; 364-366 among others.

4. Conclusions

The authors should embase  their conclusion with examples of biomarkers that can be apllayed in snakebite therapy.

Author Response

Point 1 Results

Line 128...specific to E-cEV (Figure 2A). We found that 2,455 proteins were quantified in E-vEV (Figure 2E), while 2,476 proteins were quantified in E- 129

the authors could present the results in the same sequence that they present in the figure, making the reading easier.

Response 1: We appreciate your suggestion to present the results in the same sequence as depicted in Figure 2, as it enhances the manuscript's readability. And thus, we have revised the manuscript to align the presentation of the results with the sequence in Figure 2. This modification allows for a smoother flow of information and facilitates better understanding for readers.

Point 2 Figure 2. the graphs presented in purple and yellow could be showed in different colors from those used to express the upregulated and downregulated proteins to avoid confusion in data analysis. The same pattern should be applied in figure 5 and 6 instead of using blue and red colors to represent upregulated and downregulated proteins.

Response 2: Thank you for your feedback regarding the color scheme used in our graphs. We have made the necessary modifications to the color scheme in Figure 5 and Figure 6. Hence, instead of using similar colors to those that represent upregulated and downregulated proteins, we have chosen colors visually distinct in other parts of the figures. This adjustment ensures a clearer distinction between different elements and aids in accurately interpreting the data.

Point 3 2.4.

Line 262:  The authors should explain with more refinement about the statement in relation to downregulation of exudate expression with upregulation of plasma protein expression.

Response 3: To clarify this, we amended the sentence: “Most of these changes demonstrate the downregulation of protein expression in the exudate, while the upregulation of protein expression is observed in the plasma”. The explanation with more refinement about this statement has been explicitly stated in the discussion.

Point 4 Discussion

the discussion is very extensive. I suggest that the authors synthesize this topic and contest and/or deepen the discussion of some points such as those presented in lines 309-31; 364-366 among others.

Response 4: We appreciate the reviewer's suggestion. We have revised and improved the text based on the reviewer's recommendations.

Point 5 Conclusions

The authors should embase their conclusion with examples of biomarkers that can be applied in snakebite therapy.

Response 5: Thank you for the suggestion. Snakebite envenomation biomarkers are an intricate matter. We definitely can explore the idea of mentioning some molecules that could serve as protein biomarkers; however, to be able to precisely indicate any specific biomarkers exist that are snakebite specific would require extensive secondary validation and additional assays beyond the scope of the experiments presented. We mentioned this aspect in the conclusion section. 

Round 2

Reviewer 3 Report

Excellent revision. No further comments.

Reviewer 4 Report

The authors revised the manuscript and made the necessary modifications to improve its comprehensible. Moreover, the manuscript has been revised and improved in many respects based on the recommendation of other reviewers. Thus, this work is suitable for publication in Toxins.